# Optimised *Agrobacterium*-Mediated Transformation and Application of Developmental Regulators Improve Regeneration Efficiency in Melons

**DOI:** 10.3390/genes14071432

**Published:** 2023-07-12

**Authors:** Lili Wan, Zhuanrong Wang, Xuejun Zhang, Hongxia Zeng, Jian Ren, Na Zhang, Yuhong Sun, Tang Mi

**Affiliations:** 1Institute of Crop, Wuhan Academy of Agricultural Sciences, Wuhan 430065, China; wanlili13226@163.com (L.W.);; 2Research Center of Hami Melon, Xinjiang Academy of Agricultural Sciences, Urumqi 830091, China; 3Hainan Sanya Crops Breeding Trial Center, Xinjiang Academy of Agricultural Sciences, Sanya 572019, China

**Keywords:** *Agrobacterium* infection, CRISPR/Cas9, developmental regulators, genetic transformation

## Abstract

Melon (*Cucumis melo* L.) is a protected crop in China with high economic value. *Agrobacterium*-mediated genetic transformation is a powerful tool to improve agronomic traits and obtain elite germplasm. However, current transformation protocols in melons are inefficient and highly genotype-dependent. To improve transformation in melon, we tested different infiltration methods for *Agrobacterium*-mediated transformation. Among these methods, micro-brushing and sonication for 20 s, followed by vacuum infiltration at −1.0 kPa for 90 s, resulted in the strongest green fluorescent protein signal and increased the proportion of infected explants. We transformed melon with developmental regulatory genes *AtGRF5*, *AtPLT5*, *AtBBM*, *AtWUS*, *AtWOX5*, and *AtWIND1* from *Arabidopsis* and estimated regeneration frequencies as the number of regenerating shoots/total number of inoculated explants in the selection medium. The overexpression of *AtGRF5* and *AtPLT5* in melon resulted in transformation efficiencies of 42.3% and 33% in ZHF and 45.6% and 32.9% in Z12, respectively, which were significantly higher than those of the control. *AtGRF5* and *AtPLT5* expression cassettes were added to CRISPR/Cas9 genome-editing vectors to obtain transgenic phytoene desaturase *CmPDS* knockout mutants. Using *AtGRF5* or *AtPLT5*, multi-allelic mutations were observed at *CmPDS* target sites in recalcitrant melon genotypes. This strategy enables genotype-flexible transformation and promotes precise genome modification technologies in melons.

## 1. Introduction

The family *Cucurbitaceae* includes numerous economically important vegetable and fruit species, among which melons, squash, watermelons, and cucumbers are the most widely cultivated. However, there remains potential for improvements in yield, quality, and resistance to abiotic and biotic stress by agricultural breeding and biotechnological strategies [1]. In particular, clustered regularly interspaced short palindromic repeats (CRISPR)/CRISPR-associated nuclease 9 (Cas9) technology is a powerful approach for achieving breeding goals [2]. However, its application depends on a robust and universal method of delivering CRISPR/Cas9 reagents into plant cells [3]. *Agrobacterium*- and biolistic-mediated transformation are the most widely used delivery methods for generating transgenic plants. *Agrobacterium*-mediated transformation remains the top choice to produce genome-edited germplasm in crops owing to its cost-effectiveness and capacity to transfer large DNA fragments into chromosomes. Although successful transformation and genome-editing methods have enabled rapid improvements in yield, quality, and resistance in recent years [4,5,6], further expansion of the universal transformation of recalcitrant species or genotypes remains an obstacle [7]. Complete plant regeneration from one or more plant cells via somatic embryogenesis and organogenesis is a particularly challenging step. Organogenesis refers to the direct or indirect organ generation from cultured explants. In direct organogenesis, new organs are formed directly from explant tissues; by contrast, in indirect organogenesis, organs are induced de novo from intermediate tissues, such as the callus, and undergo gradual dedifferentiation [8,9,10]. For horticultural crops, both organogenesis routes are possible. Direct generation is advantageous because it can prevent spontaneous mutations and yield regenerated plants within a shorter time frame [10]. In melon, the initial target cells capable of producing the meristem or whole plantlets are present in the deeper layers of vascular tissue, including the procambium and cambium [6,11]. During direct organogenesis, adventitious shoots can originate from the procambial or cambial cells [12]. A critical step towards achieving efficient transformation is the effective infiltration of vascular cells during direct organogenesis. To a certain extent, transformation efficiency can be enhanced by increasing the infection intensity. However, inefficient infiltration and over-infiltration of the cambium cells in the vascular tissue may lead to variable infection rates mainly due to a failure to transform regeneration-competent cells or severe explant damage. Ineffective *Agrobacterium* infection hinders its widespread application in cucurbitaceous species.

Although organs and whole plantlets can be regenerated from plant cells that are considered totipotent, natural regeneration from somatic cells is not possible for many plant species. In vitro tissue culture of various explants relies on the application of plant hormone combinations (mainly auxins and cytokinins) to enriched media to enable plantlet regeneration [13]. Establishing a successful de novo organogenesis or transformation protocol often requires customising the cytokinin-to-auxin ratio and other culture factors for each genotype. Understanding plant regeneration mechanisms is crucial for transformation, especially for species or genotypes recalcitrant to transformation. A significant breakthrough for genotype-flexible transformation would be the application of specific morphogenic factors to reprogram somatic cells to induce shoot or somatic embryogenesis [10]. This has sparked new interest in exploiting specific developmental regulators (DRs) for plant transformation. These factors include plant-specific transcription factors, such as the AP2/ERF-family transcription factor *BABY BOOM (BBM)* [14], shoot apical meristem identity regulator *WUSCHEL (WUS)* [15,16], *WUSCHEL-related homeobox (WOX)* [17], *GROWTH-REGULATING FACTOR 4 (GRF4)* [18], *WOUND-INDUCED DEDIFFERENTIATION 1 (WIND1)* [19], and AP2-family transcription factor *PLETHORA (PLT)* [20]. Moreover, they can promote growth and regeneration. The overexpression of *GRF4* and the *GRF-INTERACTING FACTOR 1* chimera was optimal for increasing the number of transformable crop species, including wheat, triticale, rice, and citrus [21].

In this study, we tested micro-brushing, sonication, and vacuum infiltration with different pressures to obtain optimised parameters for infiltration. To test the effects of DRs on melon transformation, we applied several DRs from *Arabidopsis* to the melon shoot regeneration process. In addition, we tested the co-expression of DRs using the CRISPR/Cas9 vector to establish an efficient method for obtaining gene-edited plants.

## 2. Materials and Methods

### 2.1. Vector Construction

For genetic transformation using the optimal infiltration strategy, a DNA fragment containing the CaMV35S promoter, CpYGFP_eYGFPuv (GenBank: LC217533.1) [22] coding sequence, and At5g59720 terminator were synthesised and introduced into the pCAMBIA1300 vector digested with HindIII and EcoRI, designed as a PV16 vector (Appendix A). To construct expression vectors with DRs, DNA fragments containing the coding sequences for *AtGRF5* (At3G13960), *AtPLT5* (At5g57390), *AtBBM* (At5g17430), *AtWUS* (At2g17950), *AtWOX5* (At3g11260), and *AtWIND1* (At1g78080) were amplified from the cDNA of *Arabidopsis thaliana* and confirmed via Sanger sequencing. The DR expression cassette was amplified with the primer pair and inserted into the region between the NcoI and BamHI enzyme sites in the PFGC5941 vector.

As previously described, sgRNA was inserted into the pB7_CAS9_TPC vector to obtain the genome-editing vector [23]. The binary vectors pB7_CAS9_TPC and pBS_KSgRNA were obtained from Dr. Bin Liu from the Hami-melon Research Center, Xinjiang Academy of Agricultural Sciences in China. For the assembly of the two gRNAs into pB7_CAS9_TPC, a four-primer mixture with T1-F0-PDS/T2-R0-PDS and T1-BsF-pds/T2-BsR-pds at a 1:20 ratio was used for polymerase chain reaction (PCR) amplification, along with pBS_KSgRNA and Phanta Max Super-Fidelity DNA Polymerase (Vazyme, Nanjing, China), following the manufacturer’s recommendations. PCR was conducted at 95 °C for 3 min, followed by 35 cycles of 95 °C for 15 s, 60 °C for 30 s, and 68 °C for 5 min. The T1T2 PCR product was separated on a 1.5% agarose gel and purified using a Gel Extraction Kit D2500 (OMEGA, Biel, Switzerland), according to the manufacturer’s instructions. The T1T2 PCR product was assembled into pB7_CAS9_TPC using the Golden Gate cloning method with BsaI-HFv2 and T4 ligase (NEB, Ipswich, MA, USA) [23]. A CRISPR/Cas9 vector was constructed and named pB7-CmPDS. The *AtGRF5* or *AtPLT5* expression cassettes were amplified from the pAtGRF5 and pAtPLT5 vector and inserted into the PmeI site of pB7-CmPDS to generate pB7-PDSw1 and pB7-PDSw2, respectively. Binary vectors were used to transform 5-α competent *Escherichia coli.* Positive cloning was confirmed using Sanger sequencing. Plasmids were isolated using the EZNA Plasmid Mini Kit (OMEGA) and transformed into *Agrobacterium tumefaciens* EHA105. All of the primers used are listed in Appendix A.

### 2.2. Agrobacterium-Mediated Melon Transformation

After removing the coats with a scalpel and forceps, seeds were surface-sterilised with 75% ethanol for 30 s (Appendix A). The peeled seeds were sterilised for 10 min with 2% sodium hypochlorite solution and then rinsed 4–5 times with sterile distilled water. Sterilised seeds were spread on Petri dishes containing germination medium at 28 °C in the dark for 24–36 h. Two days before transformation, an *Agrobacterium* stock solution (EHA105) carrying the expression vectors was cultured in 5 mL of liquid LB medium with 50 mg/L spectinomycin and 25 mg/L rifampicin for 16–18 h at 28 °C with gentle shaking. Starter cultures were transferred to 30 mL of liquid LB medium at a 1:1000 ratio and cultured overnight to an OD_600_ of 0.4–0.5. The *Agrobacterium* culture was centrifuged at 3000 rpm and resuspended to an OD_600_ of 0.2 with inoculation medium at 25 °C. The cotyledons were cut transversely in half, and the distal segments were removed. The proximal end of the cotyledon exhibited a U-shaped cut after removing the hypocotyl and was used as an explant for the following treatments: scratching with a micro-brush (KITA, Nanotek Brush NANO-3-003, Tokyo, Japan) near the end with the U-shaped cut, sonication using an ultrasonic cleaning instrument for 20 s, and vacuum treatment with the *Agrobacterium* suspension. For vacuum infiltration, wounded explants were placed in a 150 mL triangular glass bottle containing 30 mL of the *Agrobacterium* inoculum. Vacuum infiltration was applied for 90 s at −0.3, −0.5, or −1.0 kPa in a sterile desiccator (Fujiwara oil-free vacuum V-1500, Taizhou, China). The vacuum was released gently and slowly to avoid physical damage to the explants. After removing the excess *Agrobacterium* suspension from the surface of the explants using sterilised filter paper, infected explants were cultured in the dark in a cocultivation medium for 2 d. The explants were rinsed thrice with sterile water containing 200 mg/L timentin and then transferred to the selection medium. When explants were cultured in the selection medium for 3 d, the green fluorescent protein (GFP) intensity was measured. The explants were cultured in the selection medium for 3–4 weeks to determine the transformation effect. The green shoots regenerated from SM with 4 mg/L glufosinate-ammonium were excised and transferred to jars containing a root induction medium. The detailed compositions and sources of each tissue culture medium are listed in Appendix A.

### 2.3. Detection and Calculation of GFP Intensity

To estimate the GFP expression level in explants after cocultivation, a fluorescence microscope was used to capture images. These images were evaluated using ImageJ. A single channel was extracted by selecting the “Image-Color-Split Channels” option. The threshold was adjusted and the appropriate area was selected by choosing the “Image-Adjust-Threshold” option. To avoid errors caused by manually selecting threshold values for different images, the default threshold value was used. Note that if the image contains a scale, the threshold may need to be adjusted or the scale area may need to be removed by selecting “Edit-Fill” to avoid any potential impact on the final results. Red colour was used to mark the selected area, and “Dark Background” was selected for the background of fluorescence images. In the Auto Threshold interface, an appropriate threshold was chosen by selecting ”Image-Adjust-Auto Threshold”. “Try all” was selected in this step, and a list of all thresholds set by the algorithm was obtained by clicking “OK”. Based on the results, the “Default” algorithm was selected and parameters for measurement were set (Analyze-Set Measurements). Parameters were adjusted by selecting the mean grey value and limit to threshold under Analyze-Set Measurements. Finally, detection was accomplished by selecting the Analyze-Measure and clicking “Measure”. Mean values are the average fluorescence intensity, equal to the sum of fluorescence intensity/defined area.

### 2.4. High-Throughput Tracking of Mutations (Hi-TOM) Sequencing

Hi-TOM sequencing was used to determine the editing efficiency of the target genes in positive plants [24]. First-round PCR was performed using site-specific primers with bridging sequences of 5′-GGAGTGAGTACGGTGTGC-3′ for the forward primer and 5′-GAGTTGGATGCTGGATGG-3′ for the reverse primer at the 5′ end (Appendix A). The 10 μL reaction mix contained 50 ng of genomic DNA, 0.5 μM specific primers, and 5.0 μL of Go TaqGreen Master Mix (2×) (Vazyme). The second round of PCR to amplify 1 μL of the primary PCR product was conducted in a volume of 20 μL with a pair of markers, and 4-bp barcode tags were added to the 5′ end of the primers separately for each sample. The DNA products from the secondary amplification were sequenced using an Illumina sequencer, and the sequencing data were analysed using the Hi-TOM website (http://www.hi-tom.net/hi-tom/). The editing efficiency was calculated as the ratio of mutant reads to the total number of reads at the target site. Additionally, sequencing data were analysed to determine the types of mutations induced at the target site, such as insertions, deletions, or substitutions (Appendix A).

## 3. Results

### 3.1. Optimal Strategy for Genetic Transformation in Melon

Stable and highly efficient regeneration via organogenesis or somatic embryogenesis is the basis of successful genetic transformation [25]. We first tested the genetic transformation of the melon cultivars ZHF—the female parental line of the cultivar ‘XueMi’ (*Cucumis melo* var. *saccharinus*)—and Z12—the female parental line of the cultivar ‘Wunongqingyu’ (*Cucumis melo* L. var. *chinensis* Pangalo)—both of which are relatively recalcitrant to transformation. A GFP-based system was used to evaluate the effect of vacuum infiltration on the vascular tissue from deep layers of melon cotyledon explants. To improve the infection intensity, we used sonication and micro-brush treatments as well as different vacuum infiltration pressures to estimate the infection rate. Explants were exposed to three treatments: micro-brushing, sonication, and vacuum infiltration (−0.3, −0.5, and −1.0 kPa). Low GFP fluorescence in the infected explants was observed without vacuum infiltration. A combination of micro-brushing, sonication, and vacuum infiltration at different intensities resulted in elevated GFP abundance. Compared with those in controls, explants exposed to −1.0 kPa infiltration showed a higher GFP fluorescence intensity and an increase in the proportion of explants with infected vascular tissue (Figure 1a). Although the combination of micro-brushing, sonication, and vacuum infiltration increased the infection rate, partial explants exhibited severe tissue disruption and died 2–3 d after infection in the SM. As the vacuum infiltration intensity increased, the survival rate of explants decreased gradually (Figure 1b, Appendix A). Living transformed explants were cultured in a shoot-inducing medium containing 4 mg/L Basta for 2–3 weeks. The regeneration rate of transformed explants was significantly greater under −0.5 and −1.0 kPa than in the controls (Figure 1c). To evaluate the effect of the infiltration intensity on transformation efficiency, we divided the number of GFP-positive T_0_ plants by the number of infected explants. The transformation efficiencies for ZHF and Z12 were 42.2% and 32.5%, respectively, after treatment D (micro-brushing, 30 s sonication, and vacuum infiltration at −1.0 kPa); these values were substantially higher than that of the control (Table 1). Overall, the transformation efficiency was significantly improved when the infected explants were subjected to a combined treatment of micro-brushing, 30 s sonication, and vacuum infiltration at −1.0 kPa.

### 3.2. Overexpression of DRs Promotes Shoot Organogenesis and Increases Melon Transformation Efficiency

Somatic embryogenesis involves complex cellular reprogramming and the activation of various signalling pathways. Several molecular genetic studies have suggested that the ectopic expression of transcription factor genes induces spontaneous somatic embryogenesis. Transcription factor genes, key regulators of plant cell totipotency, are ectopically expressed and induce somatic embryogenesis without requiring exogenous plant growth regulators or stress [26,27]. Furthermore, the ectopic expression of DR genes stimulates signalling pathways that promote cell proliferation and morphogenesis during embryogenesis [17,21,25,28,29,30,31,32]. Although shoot induction directly from cotyledons via organogenesis occurs at high rates in melons, transformed cells are usually concentrated 2–3 mm away from the cut sides, where shoot formation is rare. To test the effects of DRs on melon transformation via organogenesis, we cloned several *Arabidopsis* DRs reported to enhance cell division and embryogenesis in other species (Figure 2a). As melon transformation is highly genotype-dependent, we overexpressed these DRs individually in two melon cultivars (ZHF and Z12) and investigated the effects on the transformation efficiency. These two lines showed enhanced transformability when *AtGRF5* and *AtPLT5* constructs were used in the medium with or without hormone combinations (Figure 2b). ZHF and Z12 showed average transformation efficiencies with the control construct of 1.1% and 2.2% in the MS medium without hormones and 3.7% and 4.8% in the medium with hormone combinations, respectively. By contrast, introducing the *35S: AtGRF5/AtPLT5/AtBBM/AtWUS/AtWOX5/AtWIND1* constructs into the explants resulted in higher transformation efficiencies than those of the control experiment in both treatments (no hormones and the combination of 6BA and ABA). The overexpression of *AtGRF5* and *AtPLT5* in ZHF resulted in transformation efficiencies of 20.3% and 16% in hormone-free medium and 42.3% and 33% in medium with 6BA and ABA. In Z12, the overexpression of *AtGRF5* and *AtPLT5* resulted in transformation efficiencies of 26.1% and 11%, respectively, in hormone-free medium and 45.6% and 32.9% in medium containing 6BA and ABA. Transgenic explants containing the *AtGRF5* and *AtPLT5* overexpression cassettes produced more positive shoots in co-transformation experiments, suggesting that *AtGRF5* and *AtPLT5* improve transgenic event recovery. In addition, shoot formation capacity was higher using hormone combinations compared to non-hormone cultures. The transformation of *AtBBM/AtWUS/AtWOX5/AtWIND1* in melon explants resulted in increased transformation efficiency, although the differences were not statistically significant (Figure 2b,c).

### 3.3. Application of AtGRF5 and AtPLT5 to the CRISPR/Cas9 System

Targeted mutagenesis using the CRISPR/Cas9 system is an effective breeding technology that can produce desired mutations in the target gene. In this experiment, we tested the compatibility of *AtGRF5*- and *AtPLT5*-mediated transformations using genome-editing tools in melon. *AtGRF5* and *AtPLT5* driven by CaMV35Sp were inserted into the CRISPR/Cas9 genome-editing plasmid pB7-CAS9-TPC [23], generating pB7-PDSw1 and pB7-PDSw2, respectively (Figure 3a). Two previously designed spacers (sgRNA1 and sgRNA2) targeting the melon phytoene desaturase gene *Cucumis melo* (*CmPDS)* were selected [33]. We added the effective DRs *AtGRF5* and *AtPLT5* to CRISPR/Cas9 genome-editing vectors. We obtained transgenic plants with a knockout of *CmPDS* and the reference MELO3C017772.2 in the International Cucurbit Genomics Initiative database. The resulting gene-edited vector plasmids were used to transform cotyledon explants. Upon co-expressing *AtGRF5* and *AtPLT5*, we obtained nine and five *CmPDS* mutants in the ZHF and Z12 transgenic plants, respectively. To determine the mutations introduced by the CRISPR/Cas9 system in T_0_ plants, we amplified the PCR products overlapping the target sequence for melon T_0_ plants. We analysed these by Hi-TOM. Each T_0_ plant harboured multiple mutations based on the Hi-TOM sequencing data. Plants with an editing ratio at the target site (the number of reads with target mutations divided by the total number of reads for the target site) of >10% were considered ‘edited’. Sequencing data revealed that the editing frequency at the two loci ranged from 12.5% to 33.3% in the 14 independent T_0_-edited melon plants (Figure 3b). The mutations in T_0_ plants included a 1- to 6-bp deletion or a 2-bp insertion (Figure 3c). After CRISPR/Cas9-mediated phytoene desaturase gene (*PDS*) knockout, some cotyledons regenerated albino shoots. By contrast, others regenerated green shoots and exhibited mosaicism or regenerated secondary albino shoots. Complete albino plants were transferred to the propagation medium (Figure 3d).

## 4. Discussion

Plant genetic transformation is a complex process. Numerous factors, including tissue culture conditions, transformation methods, selection process, genetic factors, and environmental conditions, affect the regeneration successes of positive or gene-edited plants. Among these challenges, the low efficiency of genetic transformation methods and the adaptability of species or genotypes to the *Agrobacterium*-mediated transformation system are important factors that determine the application of transgenic and gene-editing approaches. The infection of target regeneration-competent tissues with *Agrobacterium* is a key step in stable genetic transformation. In melon transformation, adventitious shoots derived from the target are initiated from the procambium or cambium cells during organogenesis. To improve the infection of the deeper cell layers of explants, we used combinations of micro-brushing, sonication, and vacuum infiltration to ensure absolute infection of cambium cells in the vascular tissue. After the adjustment of infiltration parameters, we ensured the sufficient infection of melon explant cells without excessive *Agrobacterium* growth. In a previous study, vacuum infiltration was applied with a syringe, relying on complicated manual operation [6], the number of explants treated at each time point was limited, and the chance of bacterial contamination was high. In our study, explants in sterile containers were conveniently treated by vacuum infiltration at −1.0 kPa to improve the transformation efficiency.

The genotype dependence in melon transformation implies that genetic transformation can only be achieved in a few varieties [34]. DR-assisted methods are the most robust and widely applicable strategy for overcoming this genotypic dependence. Numerous individual DRs or DR combinations have been used to induce shoot and somatic embryogenesis in various species [35]. Growth-regulating factors are plant-specific transcription factors with varying numbers of paralogues in plant species. GRF5 promotes organogenesis in dicot species [29]. *BBM* and *WUS* promote direct somatic embryogenesis and enhance *Agrobacterium*-mediated transformation during ectopic expression in monocot species [30]. To avoid pleiotropic deleterious effects of the constitutive expression of *BBM* and *WUS* during plant growth, strategies to express these loci under the control of a tissue-specific or auxin-inducible promoter can be used to initiate DR expression during a specific stage. Furthermore, these DR cassettes can also be entirely removed from the expression vector at the post-infection stage [36]. *WOX5*, which is expressed in the quiescent centre, is essential for shoot/root apical and stem-cell maintenance in flowering plants [37]. *Arabidopsis PLT5* can improve the in-planta transformation efficiency in snapdragon (*Antirrhinum majus*) and tomato (*Solanum lycopersicum*), likely via auxin and cytokinin biosynthesis. *PLT5* overexpression reportedly allows for the tissue culture-based transformation of recalcitrant genotypes in sweet peppers [38]. In addition to hormone-induced de novo regeneration, wounds are a primary trigger of organ regeneration. The AP2/ERF transcription factor *WIND1* promotes cell dedifferentiation and proliferation during callus formation at wound sites. The overexpression of *WIND1* and its homologues stimulates callus formation and shoot regeneration. In the present study, we observed that the frequency of explants containing transgenic shoots increased significantly when *AtGRF5* was transformed into melons. Although GRF overexpression improved transgenic cell proliferation in other crops, such as canola, soybean, and sunflower, the transformation efficiency did not always increase [29]. Target explants and phytohormones for regeneration may interact with the overexpressed *AtGRF5*, which increases the sensitivity of melon explants to cytokinins and stimulates shoot regeneration when co-expressed with cytokinin catabolic enzymes. After transformation, *PLT5* overexpression stimulates callus formation and shoot regeneration at the wound site. *PLT5* functions in acquiring cellular pluripotency, preceding shoot progenitor establishment via *WIND1* and *WUS*, which explains its strong effect on genetic transformation. Cellular dedifferentiation, reflected by callus formation, re-differentiation, and de novo shoot regeneration from calli, is critical for successful gene delivery into plant cells. Despite the improvements using DRs, challenges persist in developing reliable transformation systems in recalcitrant melon species or genotypes not mentioned herein. If these DRs function hierarchically during embryogenesis and callus formation, the DRs can be combined in an additive manner to improve regeneration efficiency. Constitutively expressed DRs may have undesired pleiotropic effects, such as severe growth defects and infertility [35]. Although we did not obtain similar results in our transgenic plants, strategies to induce expression during a specific stage and the use of temporal and spatial promoters to drive expression, with the removal of the expression cassette after infection under specific culture conditions, can avoid aberrant vegetative or reproductive growth. This method can eliminate pleiotropic effects by self-pollination or crossbreeding of genome-edited plants. Although different regeneration efficiencies can be obtained upon using various DRs, developmental plasticity is based on a combination of factors, including hormone interactions, cell cycle progression, nutrient metabolism, and injury-induced reactive oxygen species (ROS) at the wound site. We observed that overexpressing six DRs could increase the transformation efficiency with the 6BA + ABA combination over that without these hormones, indicating that the type and concentration of hormones added to the culture medium also affect genetic transformation. *AtGRF5* or *AtPLT5* had greater effects on the transformation efficiency than those of other DRs, and this could be explained by their roles in phytohormonal crosstalk, ROS homeostasis, cell cycle progression, and cell division in the overexpressing plants.

In our experiment, we used DR expression cassettes in the CRISPR/Cas9 genome editing system to obtain gene-edited melon plants. Although DRs can improve the transformation efficiency, the incidence of targeted mutations is determined by the expression levels of the sgRNAs and Cas9 nuclease. The widely used *Arabidopsis* U3 and U6 promoter and rice U6 promoter exhibit high activity in monocot and dicot plants. The application of crop codon-optimised Cas9 resulted in high mutagenesis efficiency [39]. Many endogenous Pol II and Pol III promoters, especially in fruit crops, have been discovered for the optimisation of the CRISPR/Cas9 system [39,40]. Multiple tRNA-gRNA units expressed under the control of an endogenous Pol III promoter can produce simultaneous mutations at multiple targeted genomic loci and enable a higher editing efficiency than that of the simplex editing system [41]. Additionally, improved sequence selection of gRNA spacers to generate DSBs or two nearby gRNAs generally increases the editing efficiency over that obtained by the expression of a single gRNA [42]. To develop effective and robust melon genome-editing systems, endogenous pol II and pol III promoters from melon and multi-sgRNA/Cas9 systems could be used to expand the toolbox and facilitate the application of CRISPR/Cas9 technology to obtain new germplasm resources with desirable agronomic traits.

## 5. Conclusions

We demonstrated that optimised infiltration manipulation can improve *Agrobacterium* transformation into melon explant cells. Strategies for enhancing melon transformation are moving beyond culture media and conditions. Understanding the molecular mechanisms by which key regulators contribute to the reprogramming, pluripotency acquisition, and regeneration of somatic cells can aid in identifying DRs required for transforming recalcitrant species or genotypes. *AtGRF5* and *AtPLT5* overexpression notably increased genetic transformation in two melon genotypes with or without phytohormones. These DRs could trigger somatic cell embryogenesis to accomplish efficient regeneration and transformation. Finally, with the assistance of DRs in the CRISPR/Cas9 system, gene editing was reasonably effective in melons, laying the foundation for genome editing for stable transformation. The combination of optimised infiltration manipulation, DR gene expression, and the Cas9 system offers a promising approach for accelerating genome editing in genotype-flexible transformation.

## Figures and Tables

**Figure 1 genes-14-01432-f001:**
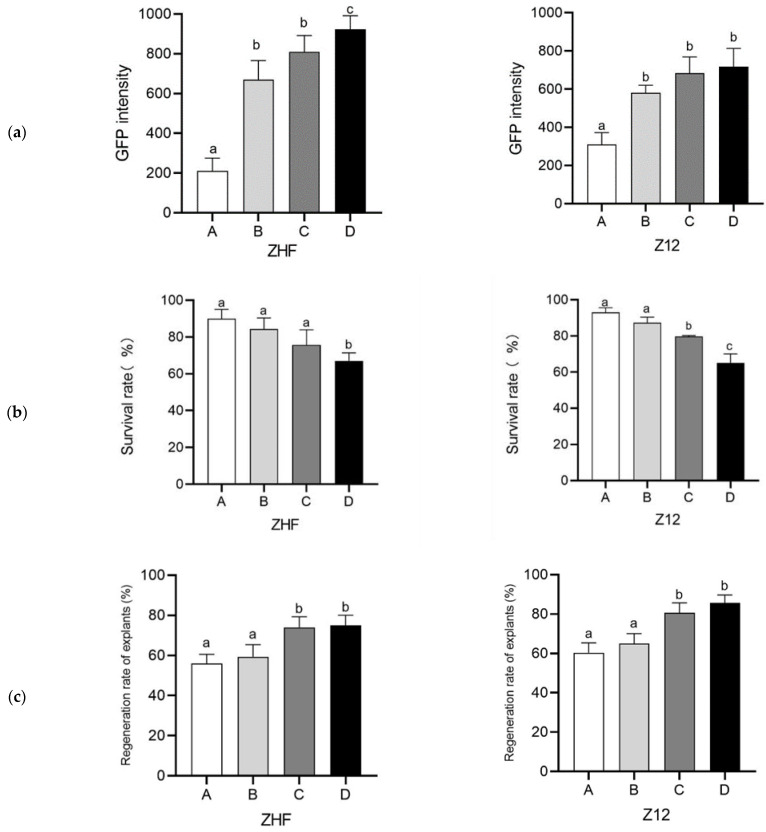
Infection of melons using different treatment approaches. (**a**) Green fluorescent protein (GFP) intensity, (**b**) survival rate, and (**c**) regeneration rate of explants with infected explant from ZHF and Z12 melon cultivars under four different treatments. Treatment A: micro-brushing (Brush) + sonication (Son) (30 s); treatment B: Brush + Son (30 s) + vacuum (−0.3 kPa); treatment C: Brush + Son (30 s) + vacuum (−0.5 kPa); treatment D: Brush + Son (30 s) + vacuum (−1.0 kPa). CK: transgenic explants without treatment. The number of explants is the sum of the surviving explants after infection in three independent experiments. Data are presented as the mean of three independent replicates, and error bars indicate standard deviations. Different lowercase letters indicate significant differences (*p* < 0.1, Tukey’s test). (**d**) GFP-fluorescent explant images were captured using a fluorescence camera. Bar = 10 mm.

**Figure 2 genes-14-01432-f002:**
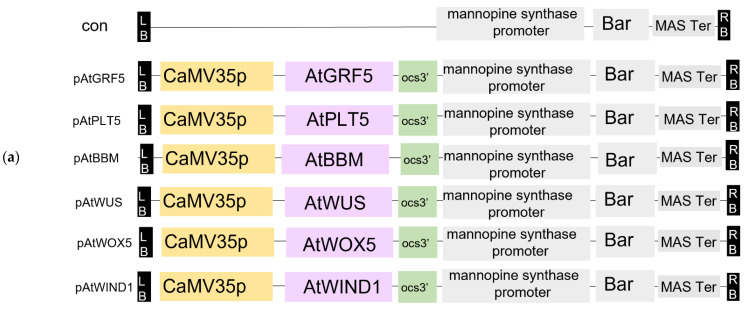
Effects of developmental regulators (DRs) on melon transformation. (**a**) Schematic diagram of constructs with DRs used in this study. (**b**) Transformation efficiencies of ZHF and Z12 melons were obtained using the indicated constructs in individual media with or without hormone combinations (6-benzylaminopurine (6BA) + abscisic acid (ABA)). Positive transformation events were defined as explants showing at least one regenerated adventitious bud expressing Basta species. Different lowercase letters indicate significant differences (*p* < 0.1, Tukey’s test). (**c**) Explants with shoot induction on selection medium with 6BA and ABA at the plant transformation stage with different DRs in ZHF and Z12, Bar = 10 mm.

**Figure 3 genes-14-01432-f003:**
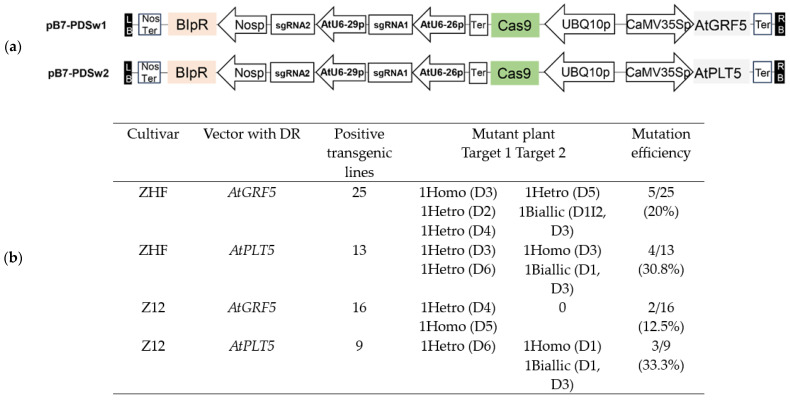
Developmental regulators (DRs) facilitate the CRISPR/Cas9-mediated genome editing of *CmPDS.* (**a**) Schematic diagrams of the CRISPR/Cas9 vector with DRs. (**b**,**c**). Mutation types at two target sites in *CmPDS* from ZHF and Z12 transformed with the *AtGRF5*- and *AtPLT5*-mediated CRISPR/Cas9 system. “D” indicates a deletion; ”I” indicates an insertion. The number indicates the length of each mutation type (bp). The PAM sequences for the two target sites are highlighted in red. (**d**) Phenotype of the regenerated CRISPR/Cas9-mutated plants. ZHF-cmpds and Z12-cmpds show fully albino plants regenerated from individual transformation. *CmPDS* shoots regenerated from transformed explants. When *CmPDS* shoots were transferred to the rooting medium, *PDS* disruption resulted in seedlings with photobleached or white leaves. Bars = 10 mm.

**Table 1 genes-14-01432-t001:** Transformation efficiency of ZHF and Z12 cultivars subjected to different treatments.

Cultivar	Treatment	No. of Explants	No. of GFP-Positive T_0_ Plants	Transformation Efficiency (%)	Transformation Efficiency ^av^ (%)
ZHF	A(rep1)	90	3	3.3%	
	A(rep2)	87	2	2.3%	3.8 ± 1.7 a
	A(rep3)	89	5	5.6%	
ZHF	B(rep1)	110	15	13.6%	
	B(rep2)	109	17	15.6%	16.7 ± 3.7 b
	B(rep3)	101	21	20.8%	
ZHF	C(rep1)	110	25	22.7%	
	C(rep2)	109	28	25.7%	22.9 ± 2.6 b
	C(rep3)	98	20	20.4%	
ZHF	D(rep1)	110	44	40.0%	
	D(rep2)	102	51	50.0%	42.4 ± 6.7 c
	D(rep3)	105	39	37.1%	
Z12	A(rep1)	88	4	4.5%	
	A(rep2)	104	5	4.8%	4.5 ± 0.4 a
	A(rep3)	98	4	4.1%	
Z12	B(rep1)	87	8	9.2%	
	B(rep2)	92	5	5.4%	6.6 ± 2.2 a
	B(rep3)	95	5	5.3%	
Z12	C(rep1)	99	19	19.2%	
	C(rep2)	101	23	22.8%	21.4 ± 1.9 b
	C(rep3)	112	25	22.3%	
	D(rep1)	109	35	32.1%	
Z12	D(rep2)	102	30	29.4%	32.5 ± 3.3 c
	D(rep3)	103	37	35.9%	

Different lowercase letters indicate significant differences (*p* < 0.05). Transformation efficiency ^av^ data are presented as the mean ± standard deviation of three independent replicates. Treatment A: micro-brushing (Brush) + sonication (Son) (30 s); treatment B: Brush + Son (30 s) + vacuum (−0.3 kPa); treatment C: Brush + Son (30 s) + vacuum (−0.5 kPa); treatment D: Brush + Son (30 s) + vacuum (−1.0 kPa).

## Data Availability

Data are contained in the paper or Appendix A.

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
