# Peer review of "Optimised Agrobacterium-Mediated Transformation and Application of Developmental Regulators Improve Regeneration Efficiency in Melons"

_genes, 2023, doi:10.3390/genes14071432_

Round 1

Reviewer 1 Report

Line 19 (abstract): delete “infection”.

Line 22 (abstract): AtGRF5, AtPLT5, AtBBM, AtWUS, AtWOX5, and AtWIND1” from the abstract it is not clear what they are…genes? Please clarify.

Line 34 (introduction): “Cucurbitaceae” please write in italics.

Line 53 (introduction): “de novo” please write in italics.

Lines 55-56 (introduction): “Direct generation is more advantageous because it can prevent spontaneous mutations and yield regenerated plants within a shorter time frame”. Please put the reference.

Lines 59-63 (introduction): “When Agrobacterium harbouring an expression vector transforms into regeneration-competent cells via a suitable operation, de novo positive shoot organogenesis commences after efficient infection. To a certain extent, transformation efficiency can be enhanced by increasing the infection intensity”. It is a confusing paragraph. Please write in a proper way.

Line 63 (introduction): “low or excessive infection” please write in a proper way because as it is too simple.

Line 68 (introduction): delete “body” and write somatic cells.

Lines 68-69 (introduction): “in vitro” please write in italics.

Line 71 (introduction): de novo please write in italics.

Lines 74-76: “A significant breakthrough for genotype-flexible transformation would be using specific morphogenic factors to reprogram somatic cells to induce shoot or somatic embryogenesis”. Please insert reference.

Line 104: Please indicate the affiliation of Prof. Bin Liu because as it is too generic.

Lines 122-134: 2.2.: In general, the paragraph named “Melon Tissue Culture Medium” is very confusing. The authors should explain better what they did; for example, they cannot write key words as “germination medium”, “inoculation medium”. And the authors should write the receipt of LB medium and they have to uniform the units of measurement.

Line 136: please write Agrobacterium in italics.

Line 146-147: the authors should prepare a picture where they can easily explain how they cut cotyledons.

I suggest to join together paragraphs 2.2 and 2.3.

Line 184: the authors mentioned somatic embryogenesis but here they used mature seeds so it is not correct to speck about somatic embryogenesis. They should speck about organogenesis. In fact, after, at the beginning of discussion, they speck about organogenesis but before on material and methods they talk about somatic embryogenesis. The text is generally very confusing.  

Table 1: why the authors did not consider the same number of explants for all the treatments? How can they compare them if the number of explants is different for each treatment?

Fig. 1 a: what does GFP “intensity” mean? How can they measure it? To me it is not clear.

Fig. 1 d: the explants are too far in all pictures. On my opinion the authors should repeat all the pictures using a lower distance and they also can capture one explant (as an example) to see clearly the GFP.

Fig. 2 c: the pictures composing the panel are too little and it is difficult to appreciate important details. The authors should show bigger pictures.

On my opinion the authors should discuss more the obtained results. In the discussion they reported different references but they did not explain a lot their results in comparison with the references reported.

Moderate editing of English language required.

Author Response

Comments and Suggestions for Authors

Response: Thank you for thoroughly evaluating the manuscript. We address the comments in the following paragraphs. The reviewer’s comments are in quotation marks and in black. 

Line 19 (abstract): delete “infection”.

Response: I have deleted infection and changed to “Agrobacterium mediated transformation”

Line 22 (abstract): AtGRF5, AtPLT5, AtBBM, AtWUS, AtWOX5, and AtWIND1” from the abstract it is not clear what they are…genes? Please clarify.

Response: The statement “AtGRF5, AtPLT5, AtBBM, AtWUS, AtWOX5, and AtWIND1” has been revised to “developmental regulatory genes AtGRF5, AtPLT5, AtBBM, AtWUS, AtWOX5, and AtWIND1 from Arabidopsis”.

Line 34 (introduction): “Cucurbitaceae” please write in italics.

Response: We have revised it.

Line 53 (introduction): “de novo” please write in italics.

Response: We have revised it.

Lines 55-56 (introduction): “Direct generation is more advantageous because it can prevent spontaneous mutations and yield regenerated plants within a shorter time frame”. Please put the reference.

Response: We have added the reference here.

Lines 59-63 (introduction): “When Agrobacterium harbouring an expression vector transforms into regeneration-competent cells via a suitable operation, de novo positive shoot organogenesis commences after efficient infection. To a certain extent, transformation efficiency can be enhanced by increasing the infection intensity”. It is a confusing paragraph. Please write in a proper way.

Response: We revised this paragraph to “A critical step towards achieving efficient transformation is effective infiltration of vascular cells during direct organogenesis. To a certain extent, transformation efficiency can be enhanced by increasing the infection intensity.”

Line 63 (introduction): “low or excessive infection” please write in a proper way because as it is too simple.

Response: We revised it to “inefficient infiltration and over-infiltration of the cambium cells in the vascular tissue.”

Line 68 (introduction): delete “body” and write somatic cells.

Response: We revised it.

Lines 68-69 (introduction): “in vitro” please write in italics.

Response: We revised it.

Line 71 (introduction): de novo please write in italics.

Response: We revised all the “de novo” in italics.

Lines 74-76: “A significant breakthrough for genotype-flexible transformation would be using specific morphogenic factors to reprogram somatic cells to induce shoot or somatic embryogenesis”. Please insert reference.

Response: We have inserted reference [10] here.

Shin, J.; Bae, S.; Seo, P.J. De Novo Shoot Organogenesis During Plant Regeneration. J Exp Bot. 2020, 71, 63-72. DOI:10.1093/jxb/erz395.

Line 104: Please indicate the affiliation of Prof. Bin Liu because as it is too generic.

Response: We revised to “Prof. Bin Liu from Hami-melon Research Center, Xinjiang Academy of Agricultural Sciences in China.”

Lines 122-134: 2.2.: In general, the paragraph named “Melon Tissue Culture Medium” is very confusing. The authors should explain better what they did; for example, they cannot write key words as “germination medium”, “inoculation medium”. And the authors should write the receipt of LB medium and they have to uniform the units of measurement.

Response: Thank you for pointing out the imprecisions in this paragraph. We revised this section to: “2.2. Tissue Culture Medium in the study. Germination medium (GM): 4.4g/L Murashige and Skoog (MS) medium (Phy-toTech LABS, M519), 30g/L sucrose, 0.5g/L Phytagel (Sigma-Aldrich, P8169), pH 5.85. Inoculation medium (IM): 1 mg/L 6-benzylaminopurine (6BA), 1 mg/L abscisic acid (ABA), 200 μM acetosyringone, and 1.25 mM morpholinoethanesulfonic acid were added to the MS medium, 30g/L sucrose, pH 5.40. Co-cultivation medium (COM): 0.5 mg/L 6BA, 1 mg/L ABA, 200 μM AS, and 1.25 mM morpholinoethanesulfonic acid were added to the MS medium, 30g/L sucrose, pH 5.40. Selection medium (SM): 1 mg/L 6BA, 0.1 mg/L ABA, 1 mg/L CuSO4·5H2O, 150 mg/L timentin, 250 mg/L cefotaxime, and 4 mg/L glufosinate-ammonium were added to the MS medium, 30g/L sucrose, pH 5.8. Root induction medium (RIM):1 mg/L indole-3-butyric acid, 150 mg/L timentin and 250 mg/L cefotaxime were added to the MS medium, 30g/L sucrose, pH 5.8. For preparing the liquid medium for Agrobacterium culture, the LB medium was supplemented with spectinomycin (50 mg/L) and rifampicin (25 mg/L). All tissue cul-tures were maintained in a culture room under a 16-h light/8-h dark cycle at 22℃ to 25℃.”

 Additionally, the abbreviation (GM, IM, COM, SM, RIM) for the medium used in this paragraph were applied in the next paragraph “2.3 Agrobacterium-mediated Melon Transformation”

Line 136: please write Agrobacterium in italics.

Response: We revised them.

Line 146-147: the authors should prepare a picture where they can easily explain how they cut cotyledons.

Response: We have provided supplementary figure 2 Explant preparation in the paper. The detailed information is listed below:

Supplemental figure 2. Explant preparation

Excision of embryos from the germinated seeds(a), we cut the cotyledons in half transversely(b), select proximal parts with U-shaped ends as explants(c).

I suggest to join together paragraphs 2.2 and 2.3.

Response: We revised sections 2.2 and 2.3. Section 2.2 “Tissue Culture Medium in the study” describes the tissue culture medium used in the study for genetic transformation experiments, while section 2.3 “Agrobacterium-mediated Melon Transformation” outlines the procedure for Agrobacterium-mediated melon transformation.

Line 184: the authors mentioned somatic embryogenesis but here they used mature seeds so it is not correct to speck about somatic embryogenesis. They should speck about organogenesis. In fact, after, at the beginning of discussion, they speck about organogenesis but before on material and methods they talk about somatic embryogenesis. The text is generally very confusing.  

Response: Sorry for the confusion. Organogenesis is the main pathway for regeneration of cotyledon explants. In Agrobacterium-mediated melon genetic transformation, Organogenesis and somatic embryogenesis could also occur via auxin and cytokinin in the medium. Therefore I revised the sentence to “Stable and highly efficient regeneration via organogenesis or somatic embryogenesis is the basis of successful genetic transformation” in 3.1. In discussion, we analyzed the effects of different DR transcription factors (GRF5, PLT5, BBM, WUS, WIND1) on promoting direct organogenesis or somatic embryogenesis of transformed explants. Among them, Arabidopsis GRF5 promotes organogenesis, PLT5 can render shoot regeneration, BBM and WUS promote somatic embryogenesis, WIND1 stimulates callus formation and shoot regeneration. All of these DR genes can promote de novo shoot regeneration, In our study, GRF5 and PLT5 significantly improved transformation efficiency in regenerated seedlings,  whether through organogenesis or somatic embryogenesis.

Table 1: why the authors did not consider the same number of explants for all the treatments? How can they compare them if the number of explants is different for each treatment?

Response: In this experiment, we used the same number of explants in each treatment. However, death occurs after excessive infection with Agrobacterium, the aborted explants cannot be included in the statistical data. The total number of explants shown in the table is the number that survived after 28-day culture on the selected medium and the regenerated positive seedlings are obtained from the surviving explants.

Fig. 1 a: what does GFP “intensity” mean? How can they measure it? To me it is not clear.

Response: In our experiment, we referred to the procedure for GFP fluorescence intensity determination in Xing Tongxu et al.,2022,9:uhab086 Horticulture Research. The detailed protocol of GFP intensity calculation are as follows:

GFP intensity determination

The protocol for calculating the GFP expression level of explants after cocultivation involves using a fluorescence microscope to capture an image. The captured image is then inserted into ImageJ, and a single channel is extracted, with segmentation of the channel being necessary if the image is stored in RGB format. In 16-bit or 8-bit format, the threshold operation can be performed directly, followed by the adjustment of the threshold and selection of the appropriate area. Users are advised to use the default value to avoid errors caused by manually selecting the threshold value for different photos. Note that if there is a scale in the picture, it is necessary to adjust the threshold or use the Edit-Fill area of the scale to remove it to avoid affecting the result. Lastly, Red can be used to characterize the selected area. When we analyze fluorescence images, dark background is selected and next the Auto Threshold interface are used to select an appropriate threshold algorithm by choosing the Try all option. All thresholds set by the algorithm will be listed, an users should choose the Default algorithm based on the result. We adjust parameters by selecting Mean gray value and Limit to threshold under Analyze-Set Measurements, and then click OK. Finally, detection can be accomplished by selecting Analyze-Measure and click “Measure”. Mean is the average fluorescence intensity.

 We added “GFP intensity determination” to “Materials and Method”.

Fig. 1 d: the explants are too far in all pictures. On my opinion the authors should repeat all the pictures using a lower distance and they also can capture one explant (as an example) to see clearly the GFP.

Response: In our experiment, the single explant expressing GFP could not be taken from sterile panel at that time in order to avoid bacterial and fungi contamination.

Fig. 2 c: the pictures composing the panel are too little and it is difficult to appreciate important details. The authors should show bigger pictures.

Response: We have revised Fig.2c to bigger pictures in the paper.

On my opinion the authors should discuss more the obtained results. In the discussion they reported different references but they did not explain a lot their results in comparison with the references reported.

Response: Thanks for your comments. In our experiment, DRs increases the transformation efficiency but not editing efficiency. We added the recent improvements of sgRNA and Cas9 in plant species in the discussion.

“In our experiment, we used DRs expression cassettes in CRISPR/Cas9 genome editing system and obtained gene-edited melon plants. Although DRs can improve the transformation efficiency, the incidence of targeted mutations are determined by the expression levels of sgRNA and Cas9 nuclease. To date, the widely used Arabidopsis U3 and U6 promoter, rice U6 promoter exhibit high activity in monocot and dicot plants. The application of crop codon-optimized Cas9 resulted in higher mutagenesis efficiency [38]. Many endogenous Pol II and Pol III promoters especially in fruit crops were discovered for the optimization of CRISPR/Cas9 system [38,39]. The multiple tRNA-gRNA units were expressed under the control of endogenous Pol III promoter can produce simultaneous mutagenesis for multiply targeted genomic loci and enables higher editing efficiency than the simplex editing system [40]. Additionally, advanced sequence selection of gRNA spacers to generate DSBs or two nearby gRNAs generally increases the editing efficiency more than the expression of a single gRNA [41]. To develop effective and robust melon genome editing systems, endogenous pol II, pol III promoters and multi-sgRNA/Cas9 systems could be used to expand the toolbox and facilitate application of CRISPR/Cas9 technology to obtain new germplasm of melon with desirable agronomic traits.”  

References

  1. Lian, Z.; Nguyen, C.D.; Liu, L.; Wang, G.; Chen, J.; Wang, S.; Yi, G.; Wilson, S.; Ozias-Akins, P.; Gong, H.; et al. Application of Developmental Regulators to Improve in Planta or In vitro Transformation in Plants. Plant Biotechnol J. 2022, 20, 1622-1635. DOI:10.1111/pbi.13837.
  2. Zhang, S.; Wu, S.; Hu, C.; Yang, Q.; Dong, T.; Sheng, O.; Deng, G.; He, W.; Dou, T.; Li, C.; et al. Increased Mutation Efficiency of Crispr/Cas9 Genome Editing in Banana by Optimized Construct. PeerJ. 2022, 10, e12664. DOI:10.7717/peerj.12664.
  3. Ren, C.; Liu, Y.; Guo, Y.; Duan, W.; Fan, P.; Li, S.; Liang, Z. Optimizing the Crispr/Cas9 System for Genome Editing in Grape by Using Grape Promoters. Hortic. Res. 2021, 8, 52. DOI:10.1038/s41438-021-00489-z.
  4. Qi, W.; Zhu, T.; Tian, Z.; Li, C.; Zhang, W.; Song, R. High-Efficiency Crispr/Cas9 Multiplex Gene Editing Using the Glycine Trna-Processing System-Based Strategy in Maize. BMC Biotech. 2016, 16, 58. DOI:10.1186/s12896-016-0289-2.

Comments on the Quality of English Language

Moderate editing of English language required.

Response: Thanks for your suggestions. We have submitted this manuscript to professional English-language editing service. The native English-speaking professionals have edited the paper for language.

Reviewer 2 Report

The manuscript titled "Optimised Agrobacterium-mediated Transformation and Application of Developmental Regulators Improve Regeneration Efficiency in Melons" presents a valuable contribution to the field of melon genetic transformation by addressing the limitations of the current protocol and proposing an improved method. The findings open up possibilities for developing elite genetic germplasm and introducing desirable agronomic traits in melons.

Major comments:

1. Can the author provide specific quantitative data to demonstrate the improvement in the transformation in the abstract?

2. Line 93: “CpYGFP_eYGFPuv (GenBank: LC217533.1)-coding 93 sequence, and At5g59720 terminator…” eYGFPuv was used here but associated references were not cited. Please cite the reference.  Reference 1, https://www.nature.com/articles/s41598-018-34837-2

3. Line 115: Figure 3a was cited before Figure 1a. Please check the all the figures and make sure that the figures are cited in order.

4. Line 146: “to an OD600 of 0.2 with the inoculation medium”. Isn’t the OD600 of 0.2 too low for the infection? Has the author test different OD or different OD been tested before? Please explain and add the related references.

5. Line 160 – 161: The GFP intensity calculation protocol was described in Xin et al.,2022 [6]. The calculation is important for the study. Please describe the calculation protocol with at least key steps as an individual subsection in section 2.

6. Line 190 – 191: To improve infection intensity, we used sonication and micro-brush treatments as well as different vacuum infiltration pressures to estimate the infection rate. Can author provide more background information about why these treatments were selected and tested? Have these treatments been tested before?

7. In table 1, Transformation efficiency data are presented as the mean ± standard deviation of three independent replicates. I tried to calculate the value of mean using No. of GFP- positive T0 plants/No. of explants. These are what I got: 10/266=0.038, 53/320=0.166, 73/317=0.230, 134/317=0.423, 5/109=0.046, 10/91=0.11, 20/101=0.198, 33/104=0.317. Is the calculation correct? Why the mean values are different with presented in the table?

8. In Figure 1d, can author add a control photo that without GFP expression?

9. In 3.1. Optimal Strategy for Genetic Transformation in Melon: What about the rooting efficiency? Can author provide a whole plant expressing GFP with a control plant? Does author genotyped the transgenic events to verify the integration of T-DNA insertion especially in shoot stage or root stage? PCR genotyping is necessary to verify the transgenic events for plant transformation study. Further experiments and validations should be conducted to confirm the stability and heritability of the introduced traits in the transformed melon plants.

10. Terminators are missing in Figure 2a, please add them.

11. A table 2 similar to the data in table 1 would be useful for readers to understand Figure 2b.

12. Add scale bar in Figure 2c and figure 3d.

13. Have the authors consider the testing of combining AtGRF5 and AtPLT5?

14. What does D3, D2, D6, D1I2 represent in Figure 3b?

15. The DNA sequence was not aligned properly in figure 3c. In general, singe “-” represents single base nucleic acid (A/T/C/G). Please reform the sequence, see example in figure 2c, figure 5, figure 6c https://academic.oup.com/plcell/article/29/6/1196/6099346?login=true

16. Provide the PCR results of Figure 3c in supplementary figure.

17. DRs increase the transformation efficiency but not editing efficiency. In the study, authors used two sgRNAs to increase the editing efficiency. In addition, the tRNA-processing system-based method improves the efficiency of CRISPR/Cas9 editing has been reported frequently in plants. I suggest author add it in discussion.  See some references: https://plantmethods.biomedcentral.com/articles/10.1186/s13007-020-00580-x

https://bmcbiotechnol.biomedcentral.com/articles/10.1186/s12896-016-0289-2

https://www.ncbi.nlm.nih.gov/pmc/articles/PMC8742547/

https://www.nature.com/articles/s41438-021-00489-z

18. Line 229-230: GFP-fluorescent explant images were captured using a fluorescence camera. Please provide a detailed description for the visualization of GFP in method section.

19. In Figure 3a, terminators are missing for all the coding sequence and sgRNAs. Please reformat AtU6-26p sgRNA1 and AtU6-26p sgRNA2 using same format of Cas9.

Minor comments:

Line 84: for increasing the number of transformable crop genotypes [21]. Can author add the associated plant species as examples?

Line 136: Please italicize Agrobacterium.

Line 137: removing the seed coats…

Line 333: add reference for “Genotype dependence in melon transformation implies that genetic transformation can only be achieved in a few varieties.”

Line 338: add reference for “Arabidopsis GRF5 promotes organogenesis in dicot species.”

Line 341-343: “Owing to the pleiotropic deleterious effects of BBM and WUS overexpression during transformation and inducible excision after the procedure, auxin/tissue-specific promoters mediated the expression of certain explants, two separate Agrobacterium strains carrying genes of interest, and the BBM/WUS cassette method [34].” This sentence is not clear to me, please rewrite it.

Line 369-370: add reference for “Sometimes, the constitutive expression of DRs may lead to undesired pleiotropic phenotypes such as severe growth defects and infertile plants.”

The quality of English language in the manuscript is generally good. However, there are a few areas where some improvements could be made to enhance readability and clarity. For example, in the sentence "Among them, micro-brushing and sonication for 20 s, followed by vacuum infiltration at -1.0 kPa for 90 s, resulted in the strongest green fluorescent protein signal and increased the proportion of infected explants," it would be helpful to rephrase the latter part of the sentence for better readability. For example, "resulted in the strongest green fluorescent protein signal and a higher proportion of infected explants.

Author Response

Comments and Suggestions for Authors

The manuscript titled "Optimised Agrobacterium-mediated Transformation and Application of Developmental Regulators Improve Regeneration Efficiency in Melons" presents a valuable contribution to the field of melon genetic transformation by addressing the limitations of the current protocol and proposing an improved method. The findings open up possibilities for developing elite genetic germplasm and introducing desirable agronomic traits in melons.

 Response: We thank this reviewer for his/her effort in thoroughly reviewing this manuscript. The detailed and constructive comments from this reviewer are much appreciated by the authors.

Major comments:

  1. Can the author provide specific quantitative data to demonstrate the improvement in the transformation in the abstract?

Response: We have provided specific quantitative data in the abstract “ Overexpression of AtGRF5 and AtPLT5 in melon with ZHF and Z12 genotypes resulted in transformation efficiency of 42.3%, 33%, 45.6%, and 32.9%, respectively, which is significantly higher than the control”.

  1. Line 93: “CpYGFP_eYGFPuv (GenBank: LC217533.1)-coding 93 sequence, and At5g59720 terminator…” eYGFPuv was used here but associated references were not cited. Please cite the reference. Reference 1, https://www.nature.com/articles/s41598-018-34837-2

Response: We have added the reference in the paper.

  1. Line 115: Figure 3a was cited before Figure 1a. Please check the all the figures and make sure that the figures are cited in order.

Response: We have checked the order of references cited in the result sections. In Figure 3a, the reference [23] have been cited in material and method “2.1 Vector Construction”, so figure 3a was cited before Figure 1a.

  1. Line 146: “to an OD600 of 0.2 with the inoculation medium”. Isn’t the OD600 of 0.2 too low for the infection? Has the author test different OD or different OD been tested before? Please explain and add the related references.

Response: In our experiment, the starter cultures were transferred to liquid LB medium and cultured overnight to OD600 0.4-0.5. The Agrobacterium culture was centrifuged and resuspended to OD600 0.2 with IM medium. This experimental method is based on Xin Tongxu et al., 2022, Targeted creation of new mutants with compact plant architecture using CRISPR/Cas9 genome editing by an optimized genetic transformation procedure in cucurbit plants. Horticulture Research.

  1. Line 160 – 161: The GFP intensity calculation protocol was described in Xin et al.,2022 [6]. The calculation is important for the study. Please describe the calculation protocol with at least key steps as an individual subsection in section 2.

Response: We have added section 2.4 GFP intensity determination in the paper.

  1. Line 190 – 191: To improve infection intensity, we used sonication and micro-brush treatments as well as different vacuum infiltration pressures to estimate the infection rate. Can author provide more background information about why these treatments were selected and tested? Have these treatments been tested before?

Response: This experimental method is based on Xin Tongxu et al., 2022, Targeted creation of new mutants with compact plant architecture using CRISPR/Cas9 genome editing by an optimized genetic transformation procedure in cucurbit plants. Horticulture Research. We referred to Xin’s methods in that paper and improve it.

  1. In table 1, Transformation efficiency data are presented as the mean ± standard deviation of three independent replicates. I tried to calculate the value of mean using No. of GFP- positive T0 plants/No. of explants. These are what I got: 10/266=0.038, 53/320=0.166, 73/317=0.230, 134/317=0.423, 5/109=0.046, 10/91=0.11, 20/101=0.198, 33/104=0.317. Is the calculation correct? Why the mean values are different with presented in the table?

Response: We have provided full data of three independent replicates in the Table1, the detailed information are listed as follow

Table 1. Transformation efficiency of ZHF and Z12 cultivars subjected to different treatments.

Cultivar

Treatment

No. of explants

No. of GFP-positive T0 plants

Transfor-mation efficiency (%)

Transformation efficiency av (%)

ZHF

A(rep1)

90

3

3.3%

A(rep2)

87

2

2.3%

3.8±1.7 a

A(rep3)

89

5

5.6%

ZHF

B(rep1)

110

15

13.6%

B(rep2)

109

17

15.6%

16.7±3.7 b

B(rep3)

101

21

20.8%

ZHF

C(rep1)

110

25

22.7%

C(rep2)

109

28

25.7%

22.9±2.6 b

C(rep3)

98

20

20.4%

ZHF

D(rep1)

110

44

40.0%

D(rep2)

102

51

50.0%

42.4±6.7 c

D(rep3)

105

39

37.1%

Z12

A(rep1)

88

4

4.5%

A(rep2)

104

5

4.8%

4.5±0.4 a

A(rep3)

98

4

4.1%

Z12

B(rep1)

87

8

9.2%

Z12          

B(rep2)

B(rep3)

C(rep1)

92

95

99

5

5

19

5.4%

5.3%

19.2%

6.6±2.2 a

Z12

C(rep2)

C(rep3)

D(rep1)

D(rep2)

D(rep3)

101

112

109

102

103

23

25

35

30

37

22.8%

22.3%

32.1%

29.4%

35.9%

21.4±1.9 b

32.5±3.3 c

Different lowercase letter indicates significant difference, (p< 0.05). Transformation efficiency av data are presented as the mean ± standard deviation of three independent replicates.

  1. In Figure 1d, can author add a control photo that without GFP expression?

Response: We have added a control photo “CK” that without GFP expression in Figure 1d.

  1. In 3.1. Optimal Strategy for Genetic Transformation in Melon: What about the rooting efficiency? Can author provide a whole plant expressing GFP with a control plant? Does author genotyped the transgenic events to verify the integration of T-DNA insertion especially in shoot stage or root stage? PCR genotyping is necessary to verify the transgenic events for plant transformation study. Further experiments and validations should be conducted to confirm the stability and heritability of the introduced traits in the transformed melon plants.

Response: In our experiments, the regenerated seedlings from explants were further screened on 4mg/L Basta. As we known, melon seedlings are sensitive to the herbicide. The survival seedlings were PCR genotyped using primers based on bar gene from genetic transformation vector. We simultaneously detected the new leaves and part of the root tissue of each individual seedling. If both can be detected, the plant was considered the positive one. The expression of CpYGFP_eYGFPuv component will hinder the continued growth of the transformed melon seedlings, and this expression component will be expressed unevenly in the plants.so we cannot provide a whole plant expressing GFP. In shoot stage and root stage, we detected Bar gene from genetic vector and confirmed the positive individual plant.

We agree with your opinion that the stability of trait inheritance needs to be determined after introducing new traits using genetic transformation.

  1. Terminators are missing in Figure 2a, please add them.

Response: We have added Terminators “MAS Ter” in Figure 2a.

  1. A table 2 similar to the data in table 1 would be useful for readers to understand Figure 2b.

Response: In Figure 2b, we showed the survival rate of explants from Treatment A, B, C and D in Figure 2b, which is more direct and accurate for the readers. The detailed data have been added in Supplementary Table S2 and Table S3.

Table S3 Survival rate of melon variety ZHF in Treatment A, B, C and D

Treatment A

ZHF

No. of survival explants

No. of explants

survival rate %

Rep 1

81

90

90

Rep 2

83

87

95

Rep 3

76

89

85

Average of survival rate %

90±5a

Treatment B

ZHF

No. of survival explants

No. of explants

survival rate %

Rep 1

94

110

85

Rep 2

85

109

78

Rep 3

91

101

90

Average of survival rate %

84±6a

Treatment C

ZHF

No. of survival explants

No. of explants

survival rate %

Rep 1

94

110

85

Rep 2

80

109

73

Rep 3

68

98

69

Average of survival rate %

75±8a

Treatment D

ZHF

No. of survival explants

No. of explants

survival rate %

Rep 1

76

110

69

Rep 2

63

102

62

Rep 3

74

105

70

Average of survival rate %

67±4b

The number of explants is the sum of the surviving explants after infection in three independent experiments. Average of survival rate (%) represents the mean of three independent replicates Different lowercase letters indicate significant differences (P<0.1, Tukey’s test).

Table S4 Survival rate of melon variety Z12 in Treatment A, B, C and D

Treatment A

Z12

No. of survival explants

No. of explants

survival rate %

Rep 1

84

88

95

Rep 2

95

104

91

Rep 3

90

98

92

Average of survival rate %

93±3a

Treatment B

Z12

No. of survival explants

No. of explants

survival rate %

Rep 1

78

87

90

Rep 2

81

92

88

Rep 3

80

95

84

Average of survival rate %

87±3a

Treatment C

Z12

No. of survival explants

No. of explants

survival rate %

Rep 1

75

99

76

Rep 2

78

101

77

Rep 3

84

112

75

Average of survival rate %

76±1b

Treatment D

Z12

No. of survival explants

No. of explants

survival rate %

Rep 1

71

109

65

Rep 2

71

102

70

Rep 3

62

103

60

Average of survival rate %

65±5c

The number of explants is the sum of the surviving explants after infection in three independent experiments. Average of survival rate (%) represents the mean of three independent replicates Different lowercase letters indicate significant differences (P<0.1, Tukey’s test).

  1. Add scale bar in Figure 2c and figure 3d.

Response: We have added scale bar in figure 2c and figure 3d.

  1. Have the authors consider the testing of combining AtGRF5 and AtPLT5?

Response: We appreciate this good idea. We will combine AtGRF5 and AtPLT5 expression cassettes in a genetic transformation vector in the following research, and test the effect of co expression of these DRs on melon regeneration through a single genetic transformation test. To avoid the potential negative effects of using the same promoter in the AtGRF5 and AtPLT5 expression components, we will test these two genes under different constitutive promoters such as CaMV35Sp and ubiquitin10p.

  1. What does D3, D2, D6, D1I2 represent in Figure 3b?

Response: We have revised it in the annotation of Figure 3b. Mutation “D”means deletion, Mutation“I”means insertion. The numbers after the letter “D”or “I” represents base pairs of deletion or insertion.

  1. The DNA sequence was not aligned properly in figure 3c. In general, singe “-” represents single base nucleic acid (A/T/C/G). Please reform the sequence, see example in figure 2c, figure 5, figure 6c https://academic.oup.com/plcell/article/29/6/1196/6099346?login=true

Response: We have revised figure 3c as shown in the following figure.

  1. Provide the PCR results of Figure 3c in supplementary figure.

Response: We have added different mutation types tracked by Hi-TOM in Table S4. The detail information are listed as follows:

Table S2 Different mutation types tracked by Hi-TOM

A01 #ZHF line1 Target 1 (Homozygous mutation)

Sort

Reads number

Ratio

Left variation type

Left variation

Right variation type

Right variation

1

2315

95%

3D

TGG

3D

TGG

C02 #ZHF line 11 Target 1(heterozygous mutation)

Sort

Reads number

Ratio

Left variation type

Left variation

Right variation type

Right variation

1

1125

52%

6D

TGGGCG

6D

TGGGCG

2

1057

48%

-

WT

-

WT

D5 #Z12 line 3 Target 1 (Homozygous mutation)

Sort

Reads number

Ratio

Left variation type

Left variation

Right variation type

Right variation

1

1891

96%

5D

GTGGG

5D

GTGGG

F2 #Z12 line 9 Target 1(heterozygous mutation)

Sort

Reads number

Ratio

Left variation type

Left variation

Right variation type

Right variation

1

1563

54%

6D

GTGGGC

6D

GTGGGC

2

1338

46%

-

WT

-

WT

B05 #ZHF line4 Target 2(heterozygous mutation)

Sort

Reads number

Ratio

Left variation type

Left variation

Right variation type

Right variation

1

1455

52%

1D

2I

A

>GC

1D

2I

A

>GC

2

1326

48%

3D

CAC

3D

CAC

C06 #Z12 line10 Target 2(heterozygous mutation)

Sort

Reads number

Ratio

Left variation type

Left variation

Right variation type

Right variation

1

1231

55%

1D

A

1D

A

2

1018

45%

3D

CAC

3D

CAC

  1. DRs increase the transformation efficiency but not editing efficiency. In the study, authors used two sgRNAs to increase the editing efficiency. In addition, the tRNA-processing system-based method improves the efficiency of CRISPR/Cas9 editing has been reported frequently in plants. I suggest author add it in discussion.  See some references: https://plantmethods.biomedcentral.com/articles/10.1186/s13007-020-00580-x

https://bmcbiotechnol.biomedcentral.com/articles/10.1186/s12896-016-0289-2

https://www.ncbi.nlm.nih.gov/pmc/articles/PMC8742547/

https://www.nature.com/articles/s41438-021-00489-z

Response: Thank you for the suggestions. We have revised the discussion and cited these references.

“In our experiment, we used DRs expression cassettes in CRISPR/Cas9 genome editing system and obtained gene-edited melon plants. Although DRs can improve the transformation efficiency, the incidence of targeted mutations are determined by the expression levels of sgRNA and Cas9 nuclease. To date, the widely used Arabidopsis U3 and U6 promoter, rice U6 promoter exhibit high activity in monocot and dicot plants. The application of crop codon-optimized Cas9 resulted in higher mutagenesis efficiency [39]. Many endogenous Pol II and Pol III promoters especially in fruit crops were discovered for the optimization of CRISPR/Cas9 system [39,40]. The multiple tRNA-gRNA units were expressed under the control of endogenous Pol III promoter can produce simultaneous mutagenesis for multiply targeted genomic loci and enables higher editing efficiency than the simplex editing system [41]. Additionally, advanced sequence selection of gRNA spacers to generate DSBs or two nearby gRNAs generally increases the editing efficiency more than the expression of a single gRNA [42]. To develop effective and robust melon genome editing systems, endogenous pol II, pol III promoters from melon and multi-sgRNA/Cas9 systems could be used to expand the toolbox and facilitate application of CRISPR/Cas9 technology to obtain new germplasm with desirable agronomic traits.”

  1. Line 229-230: GFP-fluorescent explant images were captured using a fluorescence camera. Please provide a detailed description for the visualization of GFP in method section.

Response: In our experiment, we referred to the procedure for GFP fluorescence intensity determination in Xing Tongxu et al.,2022,9:uhab086 Horticulture Research. The detailed protocol of GFP intensity calculation are as follows:

GFP intensity determination

The protocol for calculating the GFP expression level of explants after cocultivation involves using a fluorescence microscope to capture an image. The captured image is then inserted into ImageJ, and a single channel is extracted, with segmentation of the channel being necessary if the image is stored in RGB format. In 16-bit or 8-bit format, the threshold operation can be performed directly, followed by the adjustment of the threshold and selection of the appropriate area. Users are advised to use the default value to avoid errors caused by manually selecting the threshold value for different photos. Note that if there is a scale in the picture, it is necessary to adjust the threshold or use the Edit-Fill area of the scale to remove it to avoid affecting the result. Lastly, Red can be used to characterize the selected area. When we analyze fluorescence images, dark background is selected and next the Auto Threshold interface are used to select an appropriate threshold algorithm by choosing the Try all option. All thresholds set by the algorithm will be listed, an users should choose the Default algorithm based on the result. We adjust parameters by selecting Mean gray value and Limit to threshold under Analyze-Set Measurements, and then click OK. Finally, detection can be accomplished by selecting Analyze-Measure and click “Measure”. Mean is the average fluorescence intensity.

 We have added “GFP intensity determination” to “Materials and Method”.

  1. In Figure 3a, terminators are missing for all the coding sequence and sgRNAs. Please reformat AtU6-26p sgRNA1 and AtU6-26p sgRNA2 using same format of Cas9.

 Response: We have revised figure 3a according to the requirement of reviewer.

Minor comments:

Line 84: for increasing the number of transformable crop genotypes [21]. Can author add the associated plant species as examples?

Response: We have described associated plant species as examples.

“Overexpression of GRF4 and its cofactor GRF-INTERACTING FACTOR 1 chimera was found optimal for increasing the number of transformable crop species including wheat, triticale, rice and citrus [21].”

Line 136: Please italicize Agrobacterium.

 Response: OK, we have revised them.

Line 137: removing the seed coats…

Response: We have revised these sentences to “After removing the coats with a scalpel and forceps, seeds were surface-sterilised with 75% ethanol for 30 s. The peeled seeds were sterilised for 10 min with 2% sodium hypochlorite solution and then rinsed 4–5 times with sterile distilled water. Sterilised seeds were spread on Petri dishes…”

Line 333: add reference for “Genotype dependence in melon transformation implies that genetic transformation can only be achieved in a few varieties.”

Response: We have added reference [34] here.

Chovelon, V.; Restier, V.; Giovinazzo, N.; Dogimont, C.; Aarrouf, J. Histological Study of Organogenesis in Cucumis Melo L. After Genetic Transformation: Why Is It Difficult to Obtain Transgenic Plants? Plant Cell Rep. 2011, 30, 2001-2011. DOI:10.1007/s00299-011-1108-9.

Line 338: add reference for “Arabidopsis GRF5 promotes organogenesis in dicot species.”

Response: We have added reference [29] here.

Kong, J.; Martin-Ortigosa, S.; Finer, J.; Orchard, N.; Gunadi, A.; Batts, L.A.; Thakare, D.; Rush, B.; Schmitz, O.; Stuiver, M.; et al. Overexpression of the Transcription Factor Growth-Regulating Factor5 Improves Transformation of Dicot and Monocot Species. Front Plant Sci. 2020, 11, 572319. DOI:10.3389/fpls.2020.572319.

Line 341-343: “Owing to the pleiotropic deleterious effects of BBM and WUS overexpression during transformation and inducible excision after the procedure, auxin/tissue-specific promoters mediated the expression of certain explants, two separate Agrobacterium strains carrying genes of interest, and the BBM/WUS cassette method [34].” This sentence is not clear to me, please rewrite it.

Response: Sorry for the confusion. We change these sentences to “To avoid the pleiotropic deleterious effects of constitutive expression of BBM and WUS during the entire plant growth, alternative strategies to express them under the control of tissue-specific or auxin inducible promoter can initiate DR expression during a specific stage. Furthermore, these DR cassettes can also be entirely removed from the ex-pression vector at post-infection stage [36]”.

Line 369-370: add reference for “Sometimes, the constitutive expression of DRs may lead to undesired pleiotropic phenotypes such as severe growth defects and infertile plants.”

 Response: We have added reference [35] here.

Chen, Z.; Debernardi, J.M.; Dubcovsky, J.; Gallavotti, A. Recent Advances in Crop Transformation Technologies. Nat. Plants. 2022, 8, 1343-1351. DOI:10.1038/s41477-022-01295-8.

Comments on the Quality of English Language

The quality of English language in the manuscript is generally good. However, there are a few areas where some improvements could be made to enhance readability and clarity. For example, in the sentence "Among them, micro-brushing and sonication for 20 s, followed by vacuum infiltration at -1.0 kPa for 90 s, resulted in the strongest green fluorescent protein signal and increased the proportion of infected explants," it would be helpful to rephrase the latter part of the sentence for better readability. For example, "resulted in the strongest green fluorescent protein signal and a higher proportion of infected explants.

 Response: Thanks for your suggestions. We have submitted this manuscript to professional English-language editing service. The native English-speaking professionals have edited the paper for language.

Round 2

Reviewer 1 Report

The materials and methods cannot be absolutely published as they are. In the paragraph named 2.3 the authors should be mentioned the media used, instead of the related abbreviations (previously explained in the paragraph 2.2). In the paragraph named 2.2 "Tissue culture media in the study" the authors provided a list including all the media used and tested on their experiments but, on my opinion, this paragraph has to be deleted completely. The paragraph named 2.4 is too detailed; the authors should reduce in a clear and proper way. On my opinion, the pictures cannot be published as they are, in particular Fig. 1 d is not clear. In supplementary Figure 2 bar missing.     

Generally, on my opinion, the whole manuscript needs to be revised again by an English native speaker. I found for example different mistakes in the abstract or the caption of supplementary figure 2 is not written in the proper way. 

Author Response

Reviewer 1

Comments and Suggestions for Authors

The materials and methods cannot be absolutely published as they are. In the paragraph named 2.3 the authors should be mentioned the media used, instead of the related abbreviations (previously explained in the paragraph 2.2). In the paragraph named 2.2 "Tissue culture media in the study" the authors provided a list including all the media used and tested on their experiments but, on my opinion, this paragraph has to be deleted completely. The paragraph named 2.4 is too detailed; the authors should reduce in a clear and proper way. On my opinion, the pictures cannot be published as they are, in particular Fig. 1 d is not clear. In supplementary Figure 2 bar missing.    

Response: We have carefully considered your comments and revised the manuscript to improve clarity. We have removed section 2.2 “Tissue culture media in the study” of the original manuscript and have included detailed information about the media in a supplemental file. The supplemental file is referred to in revised 2.2 Agrobacterium-mediated Melon Transformation. Section 2.3 “Detection and calculation of GFP intensity” provides a detailed description of the evaluation of GFP intensity according to previously described methods; we have simplified our explanation in revised paragraph 2.4. Furthermore, we have revised the figures and added images with a resolution of 300 dpi. In supplementary Figure 2, we have added a scale bar.

Comments on the Quality of English Language

Generally, on my opinion, the whole manuscript needs to be revised again by an English native speaker. I found for example different mistakes in the abstract or the caption of supplementary figure 2 is not written in the proper way.

Response: We appreciate your expertise and attention to detail in the review process. After revising the entire manuscript to address reviewer comments, we have submitted the paper to a professional English editing service for polishing. Information for the language editing service is as follows:

Reviewer 2 Report

Thank you for addressing the comments and revising the manuscript. I am pleased to see that the authors have addressed the concerns raised during the previous review. The revisions have significantly improved the clarity and quality of the manuscript.

I believe that the manuscript is now suitable for publication with only minor revisions. I only have question about response to comment 9.

Authors stated that “The expression of CpYGFP_eYGFPuv component will hinder the continued growth of the transformed melon seedlings, and this expression component will be expressed unevenly in the plants.so we cannot provide a whole plant expressing GFP”. Uneven distribution of green fluorescence in GFP-expressing transgenic plants is a commonly observed phenomenon. However, it is uncommon that the development and growth of transgenic plants are hindered by the expression of GFP. Particularly, eYGFPuv-expressing plants like Arabidopsis and poplar showed no negative consequences (cite the reference: Expanding the application of a UV-visible reporter for transient gene expression and stable transformation in plants https://www.nature.com/articles/s41438-021-00663-3). Do the eYGFPuv-expressing plants contain any DRs? If yes, the growth defects may result from DRs (as mentioned in line 403-405). If not, it would be beneficial if the authors could provide additional evidence or explanations regarding the potential adverse impact of the eYGFPuv component in melon. In any case, I recommend including the aforementioned points in the discussion section.

Author Response

Reviewer 2

Comments and Suggestions for Authors

Thank you for addressing the comments and revising the manuscript. I am pleased to see that the authors have addressed the concerns raised during the previous review. The revisions have significantly improved the clarity and quality of the manuscript.

I believe that the manuscript is now suitable for publication with only minor revisions. I only have question about response to comment 9.

Authors stated that “The expression of CpYGFP_eYGFPuv component will hinder the continued growth of the transformed melon seedlings, and this expression component will be expressed unevenly in the plants.so we cannot provide a whole plant expressing GFP”. Uneven distribution of green fluorescence in GFP-expressing transgenic plants is a commonly observed phenomenon. However, it is uncommon that the development and growth of transgenic plants are hindered by the expression of GFP. Particularly, eYGFPuv-expressing plants like Arabidopsis and poplar showed no negative consequences (cite the reference: Expanding the application of a UV-visible reporter for transient gene expression and stable transformation in plants https://www.nature.com/articles/s41438-021-00663-3). Do the eYGFPuv-expressing plants contain any DRs? If yes, the growth defects may result from DRs (as mentioned in line 403-405). If not, it would be beneficial if the authors could provide additional evidence or explanations regarding the potential adverse impact of the eYGFPuv component in melon. In any case, I recommend including the aforementioned points in the discussion section.

Response: We appreciate your expertise and attention to detail in the review process. We have revised the manuscript to improve clarity, particularly regarding the cause of growth defects and effects of expression components. In our experiment, the eYGFPuv-expressing plants did not express any DRs. In the tissue culture period, explants transformed with the eGFPuv expression vector were selected on culture medium containing 4 mg/L basta. The regenerated shoots with eYGFPuv are shown in Figure 1 and Figure 2 below.

Figure 1 Visualization of eYGFPuv expression in melon shoots post-transformation in medium. Bar = 1 cm

Figure 2 Visualization of eYGFPuv expression in melon shoots post-transformation under UV light. Bar = 1 cm

As shown in the figures, an uneven distribution of green fluorescence in eYGFPuv-expressing transgenic plants is a commonly observed phenomenon. We sampled tissues from the same seedling with and without green fluorescence. PCR genotyping of these samples using a specific marker from the eGFPuv expression vector indicated that both had genetic transformation events. Accordingly, we identified this seedling as transgenic-positive. Although we do not have sufficient data to definitively establish the effect of eGFPuv expression on plant regeneration, the constitutive expression of the eGFPuv protein may lead to the formation of inclusion bodies in the cytoplasm of plant cells. These structures may interfere with normal cellular processes and lead to reduced plant growth and development. As reported in the literature mentioned by the reviewer, various characterisitics of eYGFPuv remain unclear, including the stability of eGFPuv mRNA and protein in transgenic plants. It is worth noting, however, that many studies have also reported the successful overexpression of GFP and other transgenes in plants without negative effects on plant growth and development. The outcome depends largely on the specific transgene, the method of expression, the plant species, and environmental conditions.
